# The GDP-Mannose Dehydrogenase of *Pseudomonas aeruginosa*: An Old and New Target to Fight against Antibiotics Resistance of Mucoid Strains

**DOI:** 10.3390/antibiotics12121649

**Published:** 2023-11-22

**Authors:** Christian Hulen

**Affiliations:** Bacterial Communication and Antimicrobial Strategies Research Unit, University of Rouen Normandy, 55 Rue Saint Germain, 27000 Evreux, France; hulen.marie@orange.fr

**Keywords:** GDP-mannose dehydrogenase, *Pseudomonas aeruginosa*, mucoid strains, enzymatic mechanism, enzyme inhibitors

## Abstract

Alginates play an important role in the resistance of mucoid strains of *Pseudomonas aeruginosa* to antibiotics, as well as their persistence by escaping the immune defense system. GDP-mannose dehydrogenase (GMD) is the key enzyme in alginate biosynthesis by catalyzing the irreversible double oxidation of GDP-mannose to GDP-mannuronate. GDP-mannose dehydrogenase purified from mucoid strains exhibits strong negative cooperativity for its substrate, the GDP-mannose, with a K_M_ of 13 µM for the site of strong affinity and 3 mM for this weak of a binding. The presence of a nucleotide strongly associated with the enzyme was detected, confirming the fact that the substrate oxidation reaction takes place in two distinct steps, with the substrate blocked on the enzyme in a half-oxidation state in the form of a hemiacetal. As the GMD polypeptide has only one site for substrate binding, our results tend to confirm the fact that the enzyme functions in a dimer form. The GDP-mannose dehydrogenase inhibition strategy that we developed a few years ago, based on the synthesis of substrate analogs, has shown its effectiveness. The addition of an alkynyl radical on carbon 6 of the mannose grafted to an amino-sulfonyl-guanosine allows, at a concentration of 0.5 mM, to inhibit GMD by 90%. As we had previously shown the effectiveness of these analogs on the sensitivity of mucoid strains of *Pseudomonas aeruginosa* to aminoglycosides, this revives the interest in the synthesis of new inhibitors of GDP-mannose dehydrogenase.

## 1. Introduction

Mucoid strains of *Pseudomonas aeruginosa* are routinely found in cystic fibrosis (CF) patients’ lungs and are associated with the severe phase of the illness, leading to irreversible lung damage, respiratory failure and chronic phase of infection [1,2]. These mucoid strains produce an alginate-like polysaccharide composed of O-acetylated D-mannuronic acid (M) and its C5′ epimer L-guluronic acid (G) moieties [3,4]. The general composition of uronic acids is either MM or MG, with an M/G ratio always exceeding 1 with a variable amount of O-acetylation [5,6].

The biosynthetic pathway leading to alginate starts with the conversion of fructose-6-P (F6P) into mannose-6-P (M6P). A further two enzymatic steps yield the intermediate GDP-mannose (GDP-Man), which is irreversibly oxidized by GDP-mannose dehydrogenase (GMD, E.C. 1.1.1.132) in GDP-mannuronate (GDP-ManA) (Figure 1).

Mannuronate is believed to be polymerized and directly transported across the inner membrane using the glycosyltransferase product of the a*lg*8 gene, in conjunction with the product of the a*lg*44 gene, a bis-(3–5)-cyclic-dimeric guanosine monophosphate (c-di-GMP) binding protein, which has been shown to be required for alginate polymerization [7,8,9]. Acetylation and epimerization steps take place in the periplasm under the concerted action of AlgI, AlgJ and AlgF for O-acetylation and AlgG to epimerization [5,6,10,11].

GMD is currently believed to be the last cytoplasmic enzyme acting in alginate biosynthesis [12,13]. In mucoid strains of *P. aeruginosa*, transcription of the *alg*D gene encoding GMD is highly activated [14]. As shown in Figure 2, three functional *alg* genes, *alg*R, *alg*Q and *alg*B, are required for complete activation of *alg*D promoter [15,16,17]. AlgR acquired functionality after phosphorylation by AlgQ and can then interact with the specific sequences upstream of the promoter [18]. AlgP acts to stabilize open structures and, in the presence of the integration host factor (IHF) and the Cyclic AMP receptor protein (CRP) ensure DNA bending [19,20,21,22].

High osmolarity of the external medium [12] and the presence of antibiotics were found to play an important role in gene activation to induce alginate biosynthesis [23].

GMD has been depicted as a 290,000 Daltons molecular weight protein [24]. Deretic et al. [14] have cloned the *alg*D gene directly downstream of the *tac* promoter in a broad host range plasmid. Isopropyl β-D-thiogalactopyranoside (IPTG) induction of cells harboring this plasmid resulted in the production of a 48,000 Daltons molecular weight polypeptide, which correlated with gene size and GMD activity. GMD was found as a polymer of six identical subunits [24]. Each sub-unit presented two different domains: the amino-terminal domain, which harbored the binding sites for GDP-mannose and β-Nicotinamide adenine dinucleotide (NAD^+^), the carboxy-terminal region, which was involved in the catalytic activity with the cysteine 238 in the active site [25,26]. More recently, Snook et al. [27] analyzed the 1.55 Å crystal structure of GMD in ternary complex with its cofactor NADH and its product GDP-mannuronic acid. The enzyme is able to form a domain-swapped dimer with two polypeptide chains contributing to each active site.

This paper presents the purification and the catalytic characterization of GMD extracted from three different mucoid strains of *P. aeruginosa*: two from our laboratory collection, which were rendered mucoid by plating on a medium with high antibiotics concentration, and one isolated from a CF patient’s lungs. The presence of the substrate associated with the enzyme in a half-oxidation step is discussed, and the effects of competitive and uncompetitive inhibitors are presented.

## 2. Results

### 2.1. Purification and Characterisation of GDP-Mannose Dehydrogenase from Mucoid Strains of P. aeruginosa

GMD was extracted as described in Materials and Methods from three different mucoid strains of *P. aeruginosa* grown in a liquid medium and shaken for 24 h at 37 °C. Ammonium limitation and high NaCl concentration in the growth medium allowed bacteria to produce alginate.

GMD-specific activity calculated after protamine sulfate precipitation and solubilization in the phosphate buffer was 30, 100, and 170 mU per mg of protein for A22 alg^+^, PAO alg^+^ and NK 125502, respectively, about four times that observed in crude extracts. Proteins were then concentrated by ammonium sulfate precipitation and submitted to gel exclusion chromatography.

Figure 3 presents the elution profile of GMD for each strain. The native form of GMD was eluted as a large peak between about 95 kDal and 45 kDal for the three strains. At the top of the peak, the proteins present an apparent molecular weight of 59,000 ± 2000 Daltons. Proteins present in the peak of activity were separated by acrylamide gel electrophoresis under denaturing conditions (SDS-PAGE) and showed the presence of a majority polypeptide of 48 kDal corresponding to the molecular weight of the product of the *alg*D gene which encodes GMD as previously described by Deretic et al. [14]. The same 48 kDal polypeptide is found by SDS-PAGE after each step of purification.

The molecular weight found for the native form of the GMD protein does not correspond to that expected for the polypeptide resulting from the expression of the gene. If we look at the dispersion of the GMD molecules during elution on gel filtration, we can conclude that the composition of the GMD extracted from the bacteria is heterogeneous.

Therefore, the ACA 34 gel has been calibrated for the GMD Stokes radius determination [28] with several proteins having known radius: Ovalbumin 31 Å, Bovine Serum Albumin 35 Å, alkalin phosphatase from *E. coli* 37 Å, including Aldolase 46 Å. If the 48 kDal polypeptide gives a globular protein after folding, its Stokes radius should be 30 Å. As we found a 34 Å radius for proteins at the top of the peak (Figure 4), we assume that GMD should be an asymmetrical protein in its native form. According to the results obtained by Snook et al. [27], during the crystallization of GMD in the presence of its cofactor and the oxidation product the GDP-mannuronate, GMD should function as a dimer. The 3D representation of the GMD polypeptide, as well as that of the dimer, shows the absence of a globular structure but rather an oblong asymmetric shape with superposition of the N-ter and C-ter domains. In the dimer, two open monomers were associated to form highly intertwined chains, which places the N-ter domain of monomer A in close association with the C-ter domain of monomer B to ensure the enzymatic activity [27].

The dispersion observed in the elution profile of GMD on the exclusion gel can, therefore, be interpreted by the fact that the protein exists in two forms in the bacterium: a monomeric form and an active dimeric form.

However, when the *alg*D gene was cloned in *P. aeruginosa* alg^−^ strain on the plasmid pVD211 [14] under the control of a tac promoter inducing a two-hundred-fold overproduction of GMD in the presence of IPTG, Roychoudhury et al. purified GMD as a hexamer of 290 kDal (6 × 48 kDal) [24]. By using purification techniques described by Roychoudhury et al. [24], we did not purify GMD from our mucoid strains of *P. aeruginosa*. When we performed step 2 of Roychoudhury’s protocol (pH5 and heat treatment of crude extracts), proteins precipitated. GMD was found as expected in the supernatant, but 70% of the activity was lost. Following step 3 of their protocol (protein precipitation with 0.45 vol. of cold acetone), we did not find GMD in the precipitate but in the supernatant. Only 4% of the activity was recovered.

We think that the GMD polypeptide association observed by Roychoudhury et al. [24] in their strain was due to gene overexpression yielding high intracellular GMD concentration. This phenomenon of self-association was previously described by Dickinson [29] with UDP-glucose dehydrogenase from bovine liver. In concentrated protein solutions, UDP-glucose dehydrogenase was aggregated in partially inactive form but was dissociated to the native form on dilution.

Figure 5 shows the evolution of GMD activity with protein concentration. When protein concentration increased, total GMD activity decreased. Maximum activity was recovered on dilution. This phenomenon was also observed by Naught et al., with GMD extracted from *E. coli* BL21 transformed with the plasmid pET-3a containing an *alg*D gene [30]. The behavior of GMD extracted from our strains was similar to that of bovine liver UDP-glucose dehydrogenase. From the results on three different mucoid strains of *P. aeruginosa*, we can deduce that GDP-mannose dehydrogenase is an asymmetrical protein that can easily self-aggregate.

### 2.2. Analysis of GMD Activity: Kinetic Parameters

Initial rates of NAD^+^ reduction by GMD were followed by variation of absorbance at 340 nm as a function of substrate concentration. First, NAD^+^ reductase activity was observed in the incubation mixture without substrate. This burst had also been observed by Naught et al. [30]. So, this activity was systemically deducted from all the initial rates before plotting.

Results presented in Figure 6 were obtained with GMD extracted from *P. aerugiosa* NK 125502 and A22 alg^+^. The Eadie–Scatchard plot, v/S versus v, yielded a biphasic curve corresponding to negative cooperativity between several binding sites for the substrate. Computer analysis of the results with a special program constructed to determine K_M_ values from the interception on the axes of the asymptotes to the curve has been used [31]. K_M_s of 13 µM and 3 mM were found for a site of strong binding and weak binding, respectively, for the two strains. These values introduced into a test program based on a non-linear regression analysis gave the theoretical curve presented in the insert Figure 6 (Δ). The experimental points were distributed very close to the theoretical points. The initial rates of NAD^+^ reduction were also plotted following the Hill expression as Log(v/Vmax − v) versus Log[S]. Figure 7 shows the sigmoid curve obtained. The h-value (a representation of the number of sites) for the low and high substrate concentrations was 2, and the curve showed a strong negative cooperativity for the binding of the substrate, respectively.

The apparent K_M_ for NAD^+^ was also determined from the double-reciprocal plot of the initial rates of GDP-mannose oxidation. The K_M_ value of 0.36 mM was of the same order of magnitude as those found by Preiss [32] for GMD from *Arthrobacter* (K_M_ = 0.26 mM) and by Roychoudhury et al. [24] for GMD from *P. aeruginosa* 8835 (K_M_ = 0.185 mM).

### 2.3. GDP-Mannose Dehydrogenase: A Nucleosido-Protein

During GMD purification, protein elution from different columns was followed by UV absorbance. GMD elution peaks did not show the normal spectrum expected for a protein. Figure 8 shows the UV absorption spectra of GMD after different steps of purification and extensive dialysis against Tris buffer 10 mM, pH 7. As a standard, the UV absorption spectrum of Bovine Serum Albumin (BSA) is shown in lower-case.

As defined by the BSA spectrum, proteins routinely present an A280/260 ratio of about 2.5. After the different purification steps of GMD, the ratio A280/260 was 0.9 for gel exclusion, 0.86 after DEAE chromatography and 0.6 after Blue-Triacryl affinity chromatography. This shift of the maximum absorbance towards 260 nm augmented while GMD purification increased, suggesting the presence of a nucleoside or a nucleotide associated with the protein. This interaction is strong enough to resist chromatographic elution and extensive dialysis.

Such a result was previously found by Dickinson [29] with UDP-glucose dehydrogenase from bovine liver, which presents a ratio A280/260 of 1.3, with a maximum absorbance at 260 nm. Acid hydrolysis of this protein released one nucleotide from each protein subunit.

To confirm the presence of nucleotide on GMD, it was extensively dialyzed against distilled water, concentrated under vacuum and dissolved in D_2_O at a concentration of 4 mg/mL. These operations have been repeated three times to eliminate all H_2_O traces. The nuclear magnetic resonance of phosphorus was then measured in a Brucker AC 200E. Figure 9 shows the spectrum obtained. A signal of high intensity appeared at −1.6 ppm with a ratio signal upon noise of 10.3. In such a protein suspension in D_2_O, a phosphorus signal may become easily unobservable by NMR techniques due to the macromolecular environment producing some quenching and shift compared to the free nucleotides. The recording of at least one phosphate signal on GMD protein suggested the presence of a nucleotide tightly bound to the enzyme.

As we have detected a natural NAD^+^ reductase activity in GMD preparations, GMD was incubated 8 h at 37 °C in glycyl-glycine buffer pH 8.5 with 1 mM NAD^+^ without substrate. A positive reaction with carbazole-borate under conditions described by Knutson and Jeanes [33] derived from the technique of Bitter and Muir [34] showed the presence of a uronic acid after the incubation of GMD. As the reaction takes place in the absence of substrate, the presence of this uronic acid can only be explained if a GDP-mannose molecule is covalently associated with the protein, most likely in a half-oxidation state.

To identify the product of this reaction, GMD was then eliminated from the incubation medium by exclusion on Sephadex G25. Soluble material eluted in the total volume was adsorbed onto a DEAE-cellulose column equilibrated in distilled water. A linear NaCl gradient was applied, and the elution profile is presented in Figure 10. Molecules absorbing UV light at 260 nm were eluted at 0.08 M NaCl; the rest of the adsorbed material characterized as NAD^+^ was eluted by washing with 1M NaCl.

Molecules eluted at 0.08 m NaCl were lyophilized and, after dissolution, in a minimal volume of phosphate buffer separated by filtration on a GF05 gel column. The elution profile presented in Figure 10 showed a large peak of 260 nm-absorbing molecules at the same elution volume as GDP and a small peak at the same volume as a nucleoside sugar. Due to the presence of a positive reaction to carbazole-borate, this small peak could correspond to GDP-mannuronate. It is possible that the freeze-drying induced the hydrolysis of the bond between the phosphate of GDP and the mannuronic acid, which would explain the weak recovery of GDP-mannuronate after gel filtration on GF05.

Molecules of the large peak were then adsorbed on DEAE-cellulose to eliminate salts and eluted with 0.03 N HCl. This fraction was separated into two parts. One was lyophilized, and the residue was solubilized in D_2_O to perform ^31^P NMR analysis. A signal at −1.6 ppm identical to the signal found with GMD was observed (Figure 11). 

The other part was also lyophilized, and a KBr crystal was made with the anhydrous residue. I.R. spectrum was recovered in order to be compared with that of GDP (Figure 12). Some identical bands of absorption between 1800–1600 cm^−1^ (absorbance of –C=N– as in purine nucleus), between 1100–1000 cm^−1^ (absorbance of –OH) and 500 cm^−1^ (absorbance of –C–C–OH in sugar) have been found.

These results suggest that a nucleotide, probably a guanosyl diphosphate, could be tightly bound to GMD. Since after incubation of GMD with NAD^+^ without the addition of substrate, a uronic acid can be detected, it can be hypothesized that a molecule of GDP-mannose is covalently bound to the enzyme in a state of half-oxidation and after NAD^+^ addition, the double oxidation reaction of mannose occurs.

### 2.4. Inhibition of GMD by Guanosine and Derivatives

GMD inhibition by guanosine and analogs was a complex phenomenon. Incubation of the enzyme with the inhibitor for 10 min at 37 °C was needed to develop maximum inhibition. Figure 13 presents the initial rates of GDP-mannose oxidation in the function of guanosine concentration plotted following Dixon’s representation, 1/v versus [I] for different substrate concentrations. Biphasic curves were observed showing a mixed inhibition: competitive inhibition for the low concentrations of guanosine with a K_I_ of about 40 µM and uncompetitive inhibition for the high concentration of inhibitor with a K_I_ of about 3 mM.

The mixed inhibition observed here was in agreement with the presence of two active sites, which seemed not to be equivalent to the active enzyme extracted from the bacteria. Guanosine was able to develop competitive inhibition with respect to substrate binding and could still bind to GMD even though GDP-mannose was already engaged in an active site.

The Hill’s plot of initial rates of GDP-mannose oxidation in the function of guanosine concentration, Log[v/v_o_-v] versus Log[I], is presented in Figure 14. For each sigmoid curve representing one substrate concentration, the h-value calculated from the tangent of the inflection point was equal to 3. It is interesting to compare this value to the results of Roychoudhury et al. [24], who found three potential binding sites for guanosine on GMD.

Due to the structural analogy between the guanosine of GDP-mannose and the adenine of NAD^+^, we can hypothesize that guanosine enters into competition with NAD^+^ for its binding to the enzyme, giving the third binding site highlighted by Hill’s representation.

Mannose, guanosine, 5′-amino-5′-deoxy-guanosine (AG), 5′-(3-hydroxypropyl)-amino-5′-guanosine (APG), tetra-O-acetyl-mannose-carbonyl-aminosulfonyl-guanosine (M5′ASG) and alcynyl-6-tetra-O-acetyl-mannose-carbonyl-amino-sulfonyl-guanosine (AM5′ASG) (Figure 15) were previously tested for their inhibitory power on alginate biosynthesis into *P. aeruginosa* mucoid strains [35]. The presence of alginate around the bacteria protects it from the effect of an antibiotic, such as tobramycin [36], allowing 60% growth in the presence of the antibiotic. These GMD inhibitors are capable of restoring sensitivity to tobramycin, inducing a growth inhibition of more than 90% at a concentration of 0.1 mM in AM5′ASG in the culture medium [37]. Our inhibitors are, therefore, capable of entering bacteria and acting on the synthesis of alginates, making the bacteria sensitive to tobramycin.

To try to explain such an effect on the resistance/sensitivity of mucoid strains to antibiotics, we checked the inhibitory capacities of our molecules on the enzymatic activity of GMD. As shown in Table 1, mannose has no inhibitory effect on GMD. This confirms that the specificity of recognition of the substrate by the enzyme is linked to the presence of GDP. As shown above, guanosine has a certain inhibiting power by competing with GDP-mannose for its binding to the enzyme. By replacing the diphosphate of GDP with an amino-carbon (5′APG) or amino-sulfonated chain (M5′ASG), we significantly increase the inhibitory capacities of GDP and GDP-mannose derivatives (Table 1). The maximum inhibition, more than 90% at 0.5 mM, is obtained when a highly reactive alkyne group is added to carbon 6 of the mannose, generating an inhibitor capable of forming a covalent bond with the enzyme at its active site.

## 3. Discussion

Results presented here show that GMD activity is borne by an asymmetrical protein capable of self-aggregation in concentrated solutions, including a rapid decrease of the enzymatic activity (Figure 5). This protein harbors a binding site for the substrate GDP-mannose and a binding site for the electron acceptor NAD^+^. This observation can be related to the structural conformation of the protein. Roychoudhury et al. [25] defined two domains on the enzyme: (i) an amino-terminal domain that contains the binding sites for NAD^+^ and GDP-mannose and (ii) a carboxyl-terminal domain that includes the cysteine 268 essential for catalysis around which the active site of the enzyme should be organized. Roychoudhury et al. have also observed a conformational change in GMD structure after binding of the substrate [25].

What is the actual form of the active enzyme? Monomer or dimer? Deretic et al. [14] deduced the amino acid arrangement in GMD polypeptide from the nucleotide sequence of the *alg*D gene. The analysis of this amino acid sequence of GMD seemed to show the presence of two potential active sites. The sequence Phe-Gly-Gly-Ser-Cys-Leu-Pro-Lys around the cysteine residue 268 presents similarities with a peptide containing active site cysteine from bovine liver UDP-glucose dehydrogenase Phe-Gly-Gly-Ser-Cys-Phe-Glu-Glu-Lys [38]. Moreover, other sequences surrounding cysteine residues 83 Asn-Val-Ser-Phe-Ile-Cys-Val-Gly-Thr-Pro-Ser-Lys on GMD also present similarities to the sequences found in the active site of two dehydrogenases, the histidinol dehydrogenase of *Escherichia coli* Tyr-Ala-Ala-Ile-Leu-Cys-Gly-Val-Glu-Asp-Val and of *Salmonella thyphimurium* Tyr-Ala-Ala-Gln-Leu-Cys-Gly-Val-Glu-Glu-Ile [39]. The Cys83 potential binding site is located in the amino-terminal domain of the enzyme, whereas the Cys 268 binding site is located in the C-ter domain at the junction with the N-ter domain. Moreover, GDP-mannose dehydrogenase and histidinol dehydrogenase belong to the same group of NAD^+^-dependent four-electron-transfer dehydrogenases, which include UDP-glucose dehydrogenase [40]. UGD and GMD seem to be mechanistically similar, using a single active site to catalyze the two-step conversion of an alcohol to the corresponding acid via a thiohemiacetal intermediate [41,42].

However, the analysis of the 3D representation of the protein with the folding of the peptide chain of the monomer sends the region containing cysteine 83 towards the outside of the molecule, while cysteine 268 is found positioned in the enzymatic cavity capable of accommodate the substrate and the cofactor [27]. There is, therefore, only one active site per polypeptide chain.

The negative cooperativity between two enzymatic sites for substrate binding, the evidence of which is shown here (Figure 6 and Figure 7), with the high difference observed between the two K_M_s (13 µM and 3 mM), should be the result of (i) the conformational change of the peptide conformation and (ii) the association of two polypeptide chains to form the active enzyme. The binding of a first molecule of GDP-mannose on one monomer would modify the structure of the dimer, leading to the first oxidation step with the reduction of one molecule of NAD^+^ and the formation of a Schiff’s base, following the model of electron transfer depicted by Ordman and Kirkwood [43] for UDP-glucose dehydrogenase and depicted by Snook et al. for GDP-mannose dehydrogenase [27]. In such a conformation, GMD should not bind another molecule of the substrate except when intracellular GDP-mannose concentration would significantly increase [44].

The kinetics of NADH formation during the course of the reaction are those expected for a reaction proceeding in two distinctly separate steps: the second oxidation step does not take place until the first reaction is essentially complete, as described by Simonart et al. with UDP-glucose dehydrogenase [45]. The reaction lasted more than two hours with a slow release of GDP-mannuronate as if: (i) the second step of oxidation is the limiting step of the reaction, (ii) during this second step, GMD should act stoichiometrically [43,44]. This slow but continuous release of the product should require a large amount of GMD inside the bacteria. The augmentation of the intracellular GMD concentration observed in mucoid strains would support this hypothesis. It has been shown that *alg*D gene transcription was highly activated by three proteins [19], two of which, AlgR and AlgP, should bind the far upstream sequences of the *alg*D promoter, as shown in Figure 2.

GMD should regulate the intracellular production of alginate precursors according to the time of alginate production during bacterial growth. In the growth medium favoring *alg* gene expression, alginates are produced in the stationary phase for more than 12 h, as previously described by Pugashetti et al. [46]. GMD should be able to bind rapidly GDP-mannose produced by the cytoplasmic enzymes of the alginate biosynthesis network but would liberate slowly GDP-mannuronate. In this case, most of the GMD protein should form a complex with the substrate in a state of half-oxidation with a covalent link to the enzyme, the Schiff’s base made between the enzyme (cys268) and the carbon 6 of the mannose residue mimicking the aldehydic intermediate. The presence of a strong phosphorus signal on the purified protein (Figure 9) confirms the presence of a nucleotide strongly bound to the enzyme.

Moreover, the fact that GMD actually presented a NAD reductase activity [24,30] should support the hypothesis of a nucleoside-sugar bound on the enzyme. When GMD was incubated in the presence of NAD^+^ without substrate, new UV-absorbing molecules were found in the reactional mixture. A positive reaction with carbazole in borate was also observed, revealing the presence of a uronic acid. Chromatography on DEAE-cellulose allowed the separation of NAD^+^ from other UV-absorbing molecules. After the concentration of these molecules and gel exclusion chromatography, a large peak of GDP was observed, identified by its elution volume and the presence of specific transmission peaks in the IR spectrum (Figure 10 and Figure 12). Moreover, a small peak with the same elution volume as that of GDP-mannuronate (or GDP-mannose) could be observed (Figure 10). The concentration step by lyophilization under acidic conditions seemed to hydrolyze the phosphate link between the uronic acid and the nucleotide conducting to the high peak of GDP observed on gel exclusion, the mannuronate being not detected by UV absorption.

All of these results confirm the mechanism of action of GMD, which resembles that of UDP-glucose dehydrogenase, two proteins belonging to the same family. This mechanism involves the formation and breakdown of thiohemiacetal and thioester intermediates. Cys268 functions as the catalytic nucleophile, which is deprotonated in the course of the first oxidation step, allowing the thiolate side chain to trap the mannose under a C6-aldehydic intermediate.

GMD is the key protein in alginate biosynthesis. As shown in Figure 1, the first steps in the transformation of fructose-6 phosphate into GDP-mannose are common to the biosynthesis chain of LPS rhamnolipids and alginates. The irreversible transformation of GDP-mannose into GDP-mannuronate by GMD starts the specific alginate synthesis pathway.

Alginates synthesized by mucoid strains of *P. aeruginosa* participate in the establishment and maintenance of a chronic infection in patients with cystic fibrosis by protecting the bacteria (i) from phagocytosis by macrophages [47], (ii) from the entry of antibiotics [48], and (iii) from the action of immunoglobulins [47]. In addition, this polysaccharide does not prevent the diffusion of bacterial toxins, which accentuate the degradation of the epithelial tissues of the respiratory tract and attack the immune system. For example, exotoxin A inactivates macrophages, and elastase inactivates complement compounds (C3 and C5A) and IgG [48].

The inhibition of alginate synthesis would allow a return to the sensitivity of the bacteria to antibiotics and the immune system. For this, inhibition of the key protein in this biosynthesis, GMD, is an interesting option that continues to arouse major interest in the synthesis of new inhibitors.

A few years ago, we developed a strategy for the synthesis of GMD inhibitors based on the structure of GDP-mannose [37]. Previous studies have shown that it was essential for the binding specificity of the inhibitor compound to preserve the guanine part [24]. To ensure better transmembrane transport of these analogs, the phosphate chain was replaced by a carbon or sulpho-carbon chain of the same length. So that the inhibitor forms a covalent bond with the enzyme, thus mimicking the transition state, an alkynyl group was added to carbon 6 of the mannose. After interaction with the enzyme, the inhibitor undergoes the first oxidation step, leading to the formation of an acetylenic ketone.

Intermediate binding in the synthesis of substrate analogs has shown their effectiveness as GMD inhibitors (Table 1). From guanosine, the addition of an increasingly long carbon chain increases the inhibitory effectiveness of the compound (5′AG, 5′APG Table 1). The maximum inhibitory effect is obtained with mannosyl-amino-sulfonyl-5′-guanosine (M5′ASG) because the presence of the alkyne radical on the C6 of mannose makes it possible to form the acetylenic ketone with GMD inhibiting the enzyme in an irreversible manner. These same products, when we tested them on mucoid strains of *P. aeruginosa*, allowed for the bacteria to return to sensitivity to tobramycin, the entry of which is blocked by alginates [37]. The presence of M5′ASG at a concentration of 0.1 mM in the *P. aeruginosa* culture medium, allowing for the production of alginate, increases the sensitivity of the bacteria to tobramycin by 55% [35].

These results support the strategy chosen to construct GMD inhibitors effective in vivo to block the synthesis of alginates and restore the sensitivity of mucoid strains to antibiotics. This strategy of synthesizing GMD inhibitors to fight against mucoid strains of *P. aeruginosa* has come back into focus. Recently, two English teams have resumed synthesizing *P. aeruginosa* GMD inhibitors by manufacturing 6-thio, 6-Chloro and 6-amino-mannose derivatives of GDP-mannose. These molecules, as well as C6-methyl-GDP mannose, tend to behave like competitive inhibitors [49,50,51]. As it seems extremely difficult to eradicate persistent bacteria such as *P. aeruginosa*, which is capable of inducing chronic infections in many hospitalized patients, a strategy involving several targets, including the inhibition of alginate biosynthesis in mucoid strains, is more likely to work to reduce the proliferation of these pathogenic strains, hence the interest in the synthesis of new molecules capable of acting in vivo on specific targets.

## 4. Materials and Methods

### 4.1. Bacterial Strains and Growth Medium

*Pseudomonas aeruginosa* A22 from the Institut Pasteur collection was rendered mucoid (strain A22 alg^+^) by resistance to high concentrations of carbenicillin (800 µg/mL) as described by Piggot et al. [52].

PAO alg^+^ was selected from *P. aeruginosa*, strain Institut Pasteur 59–33, by resistance to high concentrations of kanamycin (1 mg/mL) as previously described by Deretic et al. [13].

*P. aeruginosa* mucoid strain NK 125502 was isolated from the lungs of a CF patient’s lungs in Hôpital Necker, Paris, and was a gift, many years ago, of Dr. Berche.

These three strains were grown in an ammonium-limited liquid medium complemented with glucose to stimulate alginate production (alg-glu medium). This medium containing 6 g yeast extract per liter, 10.7 g Na_2_HPO_4_ 12 H_2_O, 2.6 g KH_2_PO_4_, 10 g NaCl, 0.24 g MgSO_4_ 7H_2_O, and 0.15 g CaCl_2_ 2H_2_O, was adjusted to pH7 and supplemented with 2% (*w*/*v*) glucose. Cultures were performed at 37 °C with vigorous shaking for at least 24 h. The mucoid characters were conserved by plating the strains on agar ALG-GLU medium containing 400 µg/mL carbenicillin for A22 alg^+^ and NK 125502, 0.8 mg/mL kanamycin for PAO alg^+^.

### 4.2. Chemicals

All chemicals used in this study were of research grade. Carbenicillin, kanamycin, lysozyme, DNase, protamine sulfate, glutathione, lysozyme, NAD^+^, GDP-mannose, GDP, mannose and guanosine were obtained from Sigma-Aldrich (Saint-Quentin-Fallavier, France).

### 4.3. GDP-Mannose Dehydrogenase Purification

Bacteria grown 24 h at 37 °C on ALG-GLU medium were harvested by centrifugation for 30 min at 8000× *g*. They were washed twice with Tris-HCl 0.1 M, pH 8 and resuspended in the same buffer at about 0.2 g (wet weight) per mL.

Lysozyme was added to a final concentration of 100 µg/mL, and bacteria were incubated for a few minutes at room temperature. Gentle lysis was performed under slow agitation on ice by adding EDTA pH8 to a final concentration of 1 mM with A22 alg^+^, 3 mM with PAO alg^+^ and 10 mM with NK 125502.

After complete lysis of bacteria, a magnesium salt was added at a concentration twice that of EDTA and bacterial DNA was digested with DNase at a concentration of 100 µg/mL for 5 min at 37 °C. The lysates were centrifuged for 30 min at 30,000× *g* and 4 °C, and the cytoplasmic fractions were collected.

GMD was extracted from the cytoplasm following the technique previously described by Preiss with *Arthrobacter* [53]. Proteins were precipitated with 1% protamine sulfate, and after centrifugation, the pellet was disrupted and washed twice in phosphate buffer 0.3M, pH7. GMD was selectively solubilized from the precipitate by the phosphate buffer. Proteins were then concentrated by precipitation with 95% ammonium sulfate.

After dialysis of one hour against Tris-HCl 50 mM, pH7, MgSO_4_ 1 mM, and glutathione 20 mM, proteins were layered onto a gel exclusion column (1.6 × 120 cm) of IBF ACA 34 (IBF, Les Ulis, France), equilibrated in the same buffer plus (NH_4_)_2_SO_4_ 1M. Elution was made rapidly at 75 mL per hour to prevent GMD inactivation. Absorbance at 280 nm was continuously monitored in the effluent, and GMD activity was detected at 340 nm by reduction of NAD^+^ in the presence of GDP-mannose.

Purification was continued by ion exchange chromatography onto DEAE-Trisacryl (IBF, Les Ulis, France) and affinity chromatography for dehydrogenases on Blue-Trisacryl (IBF, Les Ulis, France). GMD elution was followed by its NAD-reduction activity. SDS-PAGE analysis shows the presence of the expected 48 kDal polypeptides previously associated with GMD [14].

### 4.4. Determination of GMD Activity

GMD activity was measured in glycyl-glycine buffer 0.2 M, pH 8.5, MgSO_4_ 10 mM, NAD^+^ 1 mM. GDP-mannose was added at various concentrations to the reactional mixture, and kinetic was started by enzyme addition. NADH formation was continuously monitored at 340 nm in a Kontron Uvikon 810 (Kontron SA, Toulon, France) spectrophotometer with the thermostat at 37 °C. One unit of activity was the appearance of one µmole of NADH per minute.

Initial rates were calculated from the slope of the kinetics and plotted against substrate concentration or as needed, against protein amounts. K_M_ value was calculated with a program developed from a least-squares non-linear regression based on the Marquardt algorithm [31].

The reaction product, mannuronic acid, is detected using the Disches’ technique modified by Bitter and Muir [34]. The presence of uronic acid is detected by reaction with carbazole in the presence of borate, making the reaction twice as sensitive [33].

Inhibition of GMD was performed in the glycyl-glycine buffer with MgSO_4_. The reaction was started after ten minutes of incubation of GMD at 37 °C in the presence of variable amounts of inhibitors by addition of substrate (variable concentrations) and 1mM NAD^+^. NADH formation was continuously monitored at 340 nm.

### 4.5. Phosphorus Nuclear Magnetic Resonance

GMD eluted from the Blue-trisacryl column was extensively dialyzed against distilled water and lyophilized. The lyophilizate was dissolved in D_2_O and lyophilized again. This operation was repeated three times to eliminate all traces of H_2_O.

About 4 mg of GMD from *P. aeruginosa* were dissolved in D_2_O and put into a quartz tube before the NMR of 31phosphorus was measured in a Brucker AC 200E (Bruker, Bremen, Germany). Phosphoric acid resonance was used as an internal standard.

### 4.6. Pufication and Analysis of GMD Associated Nucleotide

GMD was incubated in a glycyl-glycine buffer 0.2 M, pH 8.5, and MgSO_4_ 10 mM for about 8 h at 37 °C in the presence of 1 mM NAD^+^. Substrate, GDP-mannose was not added. After incubation, proteins were eliminated by Sephadex G25 gel exclusion in distilled water. UV-absorbing material (260 nm) eluted in the void volume of the column was adsorbed onto a DEAE-cellulose column equilibrated in distilled water. After washing, the adsorbed compounds were eluted with a linear gradient of NaCl, 0 to 0.1 M.

The column was then rinsed with NaCl 1 M to wash out all the adsorbed material. The 260 nm-absorbing molecules eluted at 0.08 M NaCl were concentrated by lyophilization, and after solubilization in the minimal volume of phosphate buffer 0.1 M pH 7.4, NaCl 0.5 M, they were passed through a GF05 (Merck) gel filtration column (136 × 1.6 cm, filtration rate 14 mL/h). Elution of the molecules was followed by monitoring absorbance at 260 nm. GF05 gel was calibrated with NAD^+^, GDP-mannose, GDP and guanosine with a standard molecular weight.

The compound eluted in the major peak from GF05 gel exclusion was largely diluted in distilled water and adsorbed onto a DEAE-cellulose column to eliminate phosphate and NaCl. After washing, the adsorbed molecules were eluted with 0.03 N HCl, and then lyophilized. The anhydrous residue was used to make a crystal with KBr to perform I.R. spectrum analysis. In order to compare the bands of absorbance, KBr crystals were made with GDP and GDP-mannose. Their I.R. spectra were monitored.

### 4.7. Synthesis of GMD Inhibitors

Our strategy was based on the synthesis of GDP-mannose analogs as competitive (5′AG, 5′APG and M5′ASG) and suicide inhibitors that bind irreversibly to the active site of the enzyme by covalent bonding of a reaction group present on carbon 6 of mannose (AM5′ASG) [37].

To facilitate the transport of these inhibitors inside the bacterium, the two phosphates in GDP have been replaced by a 5′-amino-sulfone group to render them more liposoluble. Their syntheses have been described by Eloumi et al. [37]. Inhibitors tested in this study are shown in Figure 15.

## Figures and Tables

**Figure 1 antibiotics-12-01649-f001:**
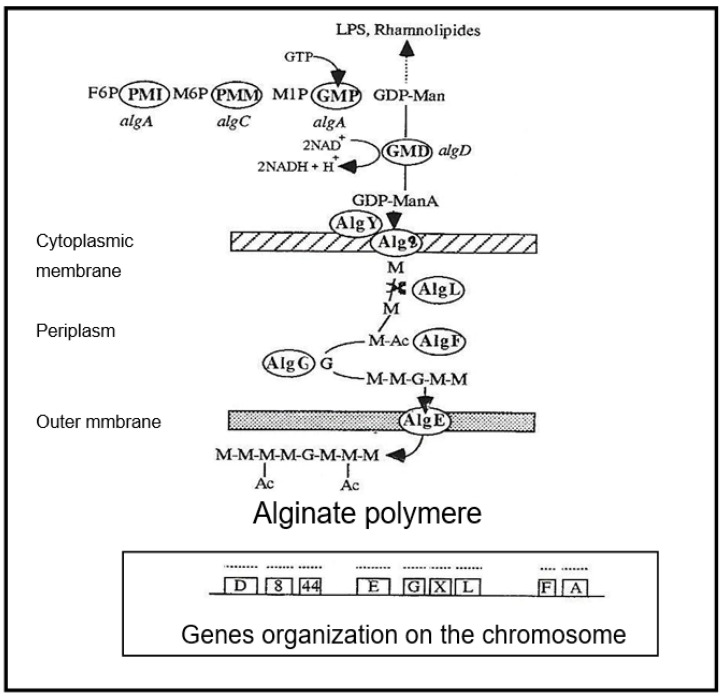
Synthetic scheme of alginate biosynthesis in mucoid strains of *Pseudomonas aeruginosa*. PMI—phosphomannose isomerase (gene *alg*A), PMM—phosphomannose mutase (gene *alg*C), GMP—GDP-mannose pyrophosphorylase (gene *alg*A), GMD—GDP-mannose dehydrogenase (gene *alg*D), AlgE—porine, AlgF—acetylase, AlgG—epimerase, AlgL—alginate lyase, Alg8—glycosyltransferase, AlgY—polymerase, *alg*44—glycosyl transferase gene, M—mannuronic acid, G—Guluronic acid, Ac—Acetylation, GTP—Guanosine triphosphate, NAD—Nicotinamide adenine dinucleotide, LPS—Lipopolysaccharide.

**Figure 2 antibiotics-12-01649-f002:**
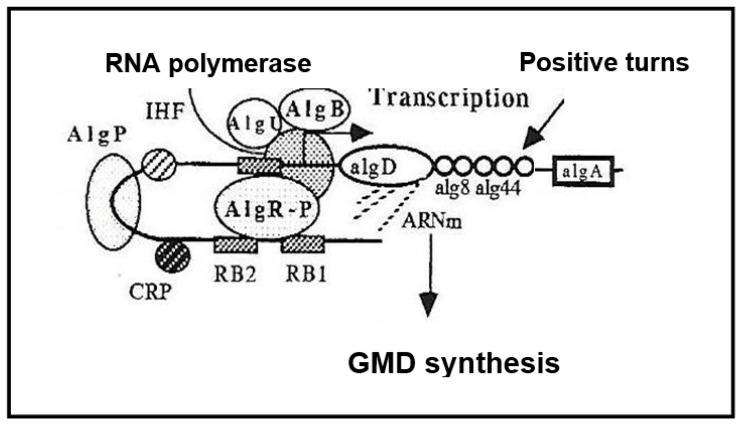
Transcriptional activation of the *alg*D gene. AlgB—transcriptional activator, AlgU—sigma-like protein, AlgP—Histone-like protein, AlgR-P—phosphorylated transcriptional activator, IHF—Integration Host Factor, CRP—cAMP receptor protein, RB—AlgR binding sites.

**Figure 3 antibiotics-12-01649-f003:**
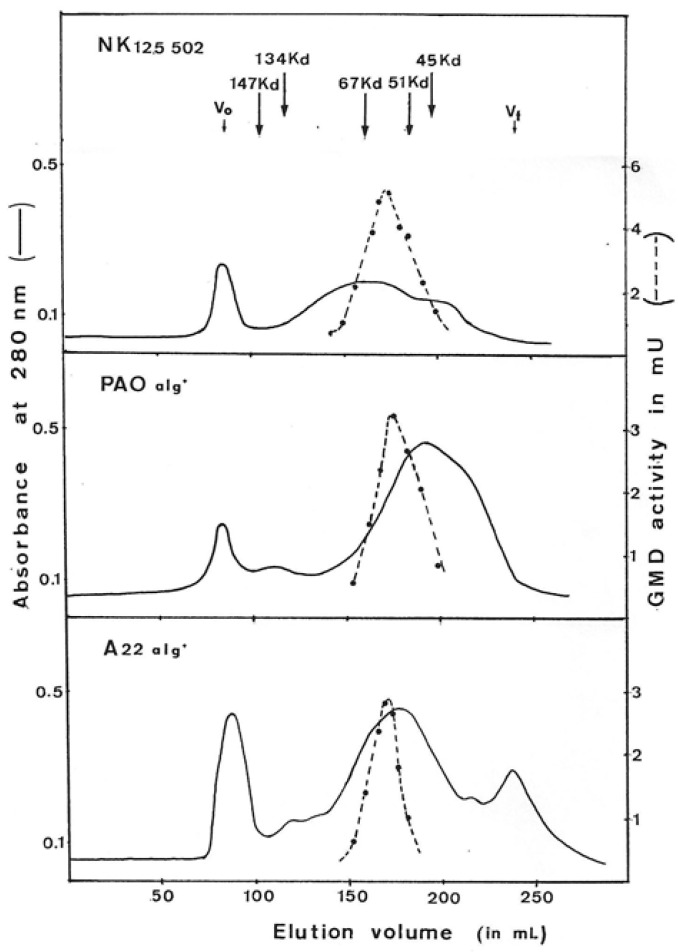
Elution profile of GDP-mannose dehydrogenase on gel filtration chromatography. GMD was prepared from three different mucoid strains of *P. aeruginosa* (NK125502, PAO alg^+^, A22 alg^+^). Ammonium sulfate-concentrated proteins obtained after selective protamine sulfate precipitation were layered onto a column of ACA (IBF) in Tris-HCL 50 mM, pH7, MgSO_4_ 1 mM, glutathione 20 mM, and (NH4)_2_SO_4_ 1M. Protein exclusion was continuously monitored by variation of absorbance at 280 nm. GMD activity was measured as described in Materials and Methods following NADH formation. V_0_—void volume, V_t_—total volume. The arrows indicate the elution volume of the standard proteins.

**Figure 4 antibiotics-12-01649-f004:**
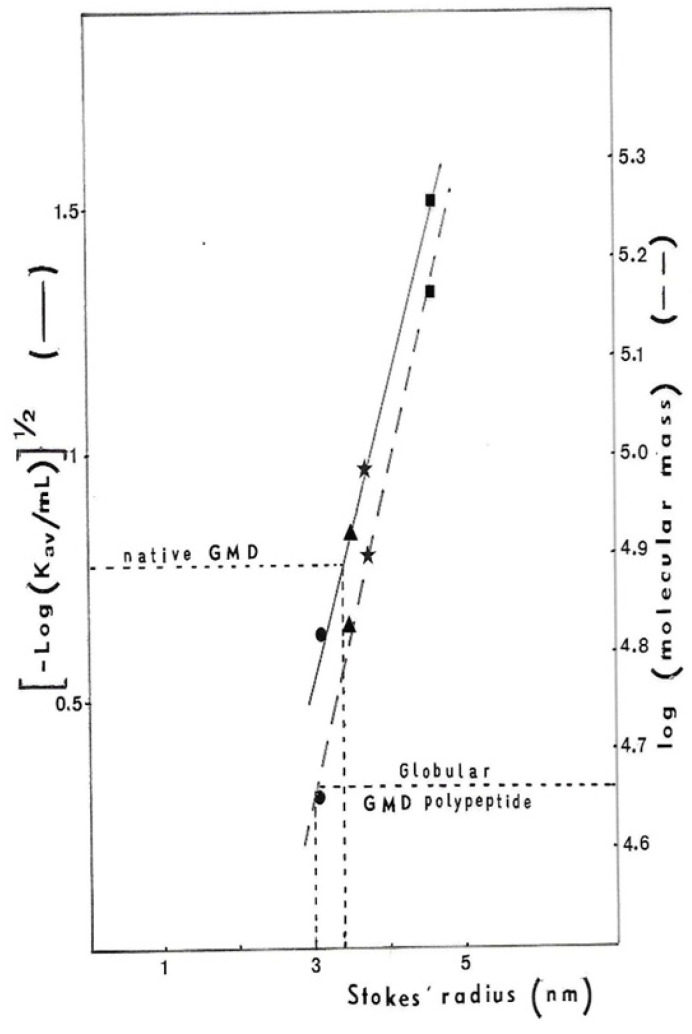
Determination of *P. aeruginosa* NK125502 GMD Stokes radius. The ACA34 gel column was calibrated for Stokes radius determination with standard proteins: ● ovalbumin 31 Å, 
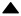
 bovine serum albumin 35 Å, 
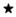

*E. coli* alkalin phosphatise 37 Å and 
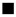
 aldolase 46 Å. The Stokes radius was plotted against either the elution volume of proteins [(−log KaV)^1/2^] or globular protein molecular weight (log MW).

**Figure 5 antibiotics-12-01649-f005:**
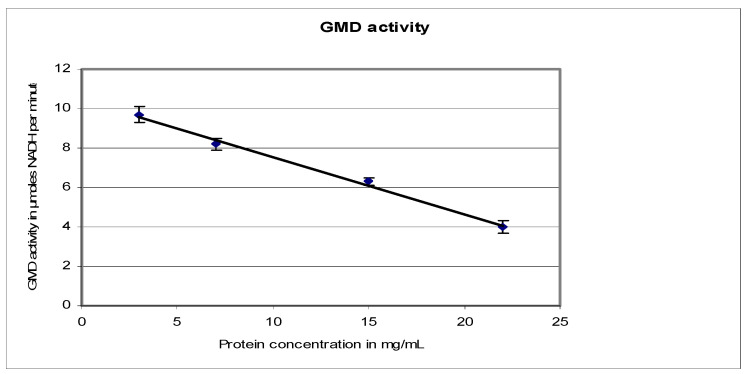
*P. aeruginosa* NK125502 GMD activity as a function of protein concentration. GMD activity was measured by following NADH formation at 340 nm in function of time. It was plotted against the variation in protein concentration in the reaction mixture.

**Figure 6 antibiotics-12-01649-f006:**
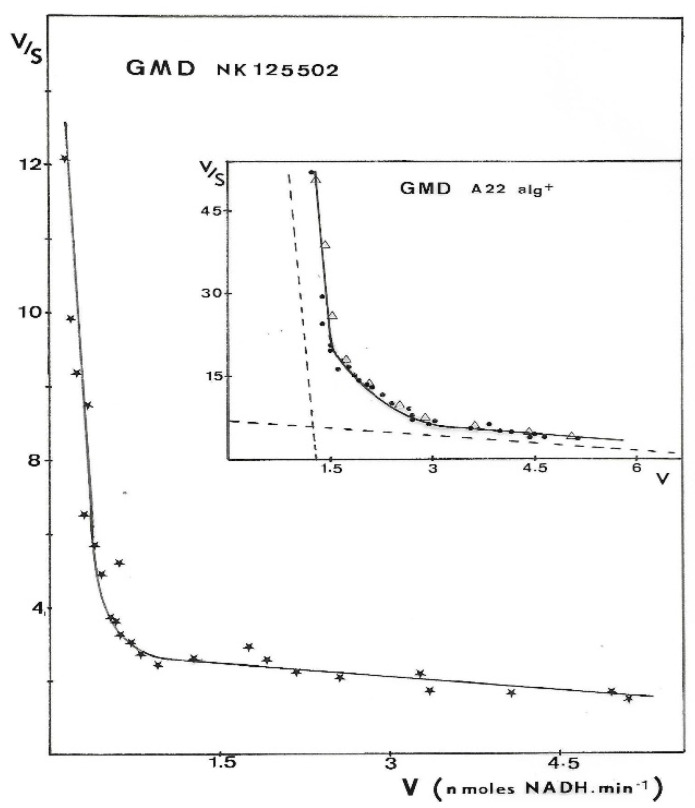
Eadie–Scatchard plot of GMD activity. GMD was purified from different mucoid strains of *P. aeruginosa*. Enzyme activity was measured as described in Materials and Methods, and initial rates of NADH formation (v) were plotted versus v/S following Eadie–Scatchard’s equation. Experimental points presented in the figure for the two strains were the result of three independent experiments with eight different concentrations in the substrate. 
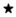
 —GMD from NK 125502, ●—GMD from A22 alg+, and Δ—theoretical curve constructed from a test program based on a non-linear regression by introducing KM values determined experimentally. Dashed lines in the insert point out the asymptotes to the curve.

**Figure 7 antibiotics-12-01649-f007:**
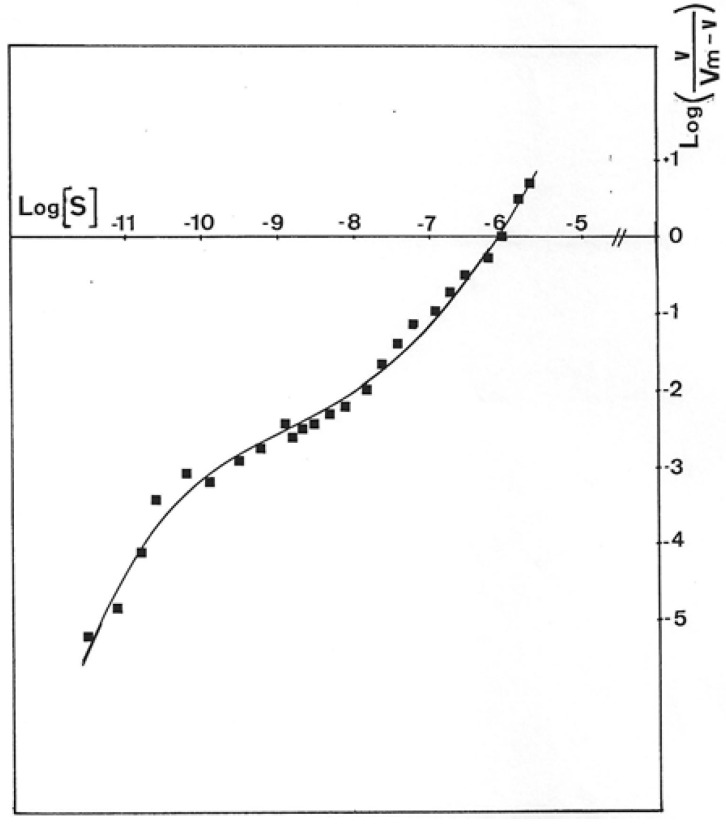
Hill plot of *P. aeruginosa* NK125502 GMD activity. Initial rates for NADH formation in the reaction mixture obtained with GMD from *P. aeruginosa* NK125502 were plotted following the representation of Hill’s equation, Log(S) versus Log(v/Vm-v). The experimental points presented in the figure were the result of three independent experiments with eight different concentrations in the substrate.

**Figure 8 antibiotics-12-01649-f008:**
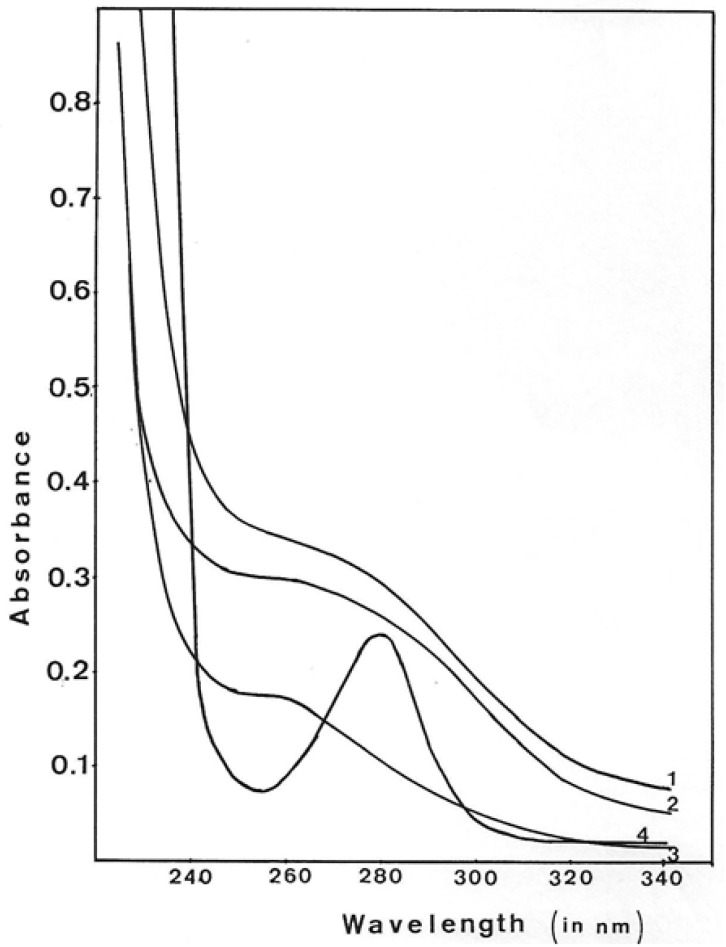
Ultraviolet spectra of GMD peaks during protein purification. The fraction with maximum GMD activity on each step of purification was dialyzed extensively against Tris-HCl 1 mM, pH 7.5, and MgSO_4_ 1 mM, and the UV spectrum was monitored. UV absorbance of GMD after 1—ACA 34 gel exclusion, 2—DEAE-Trisacryl elution, 3—Blue-Triacryl elution, 4—UV spectrum of Bovine Serum Albumin.

**Figure 9 antibiotics-12-01649-f009:**
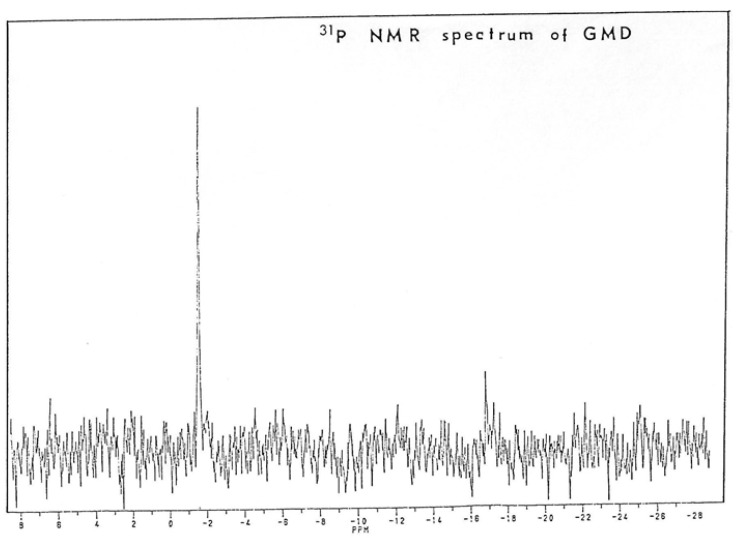
Analysis of Phosphorus Nuclear Magnetic Resonance of *P. aeruginosa* NK125502 GMD. GMD was collected after Blue-Trisacryl affinity chromatography and extensively dialyzed against water before water was exchanged with D_2_O. Proteins were lyophilized, resuspended in D_2_O, and analyzed for phosphorus NMR in a Brucker 200E. The apparatus was calibrated with phosphoric acid before recording.

**Figure 10 antibiotics-12-01649-f010:**
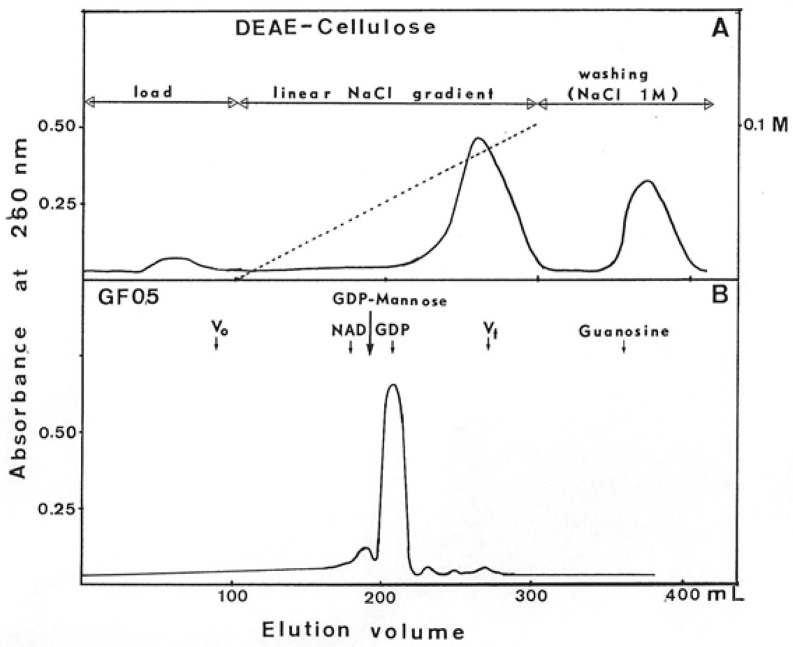
Purification of UV-absorbing molecules released by GDP-mannose dehydrogenase. After incubation 8 h at 37 °C of GMD in the presence of 1 mM NAD^+^ without substrate, UV 260 nm-absorbing molecules were separated onto DEAE-cellulose chromatography (Panel (**A**)). Molecules eluted at 0.08 M NaCl were concentrated and layered onto a GF05 gel exclusion column. Panel (**B**) presents the elution profile obtained. NAD^+^, GDP, GDP-mannose, and guanosine were used as molecular weight standards. Guanosine was unexpectedly retained on this gel.

**Figure 11 antibiotics-12-01649-f011:**
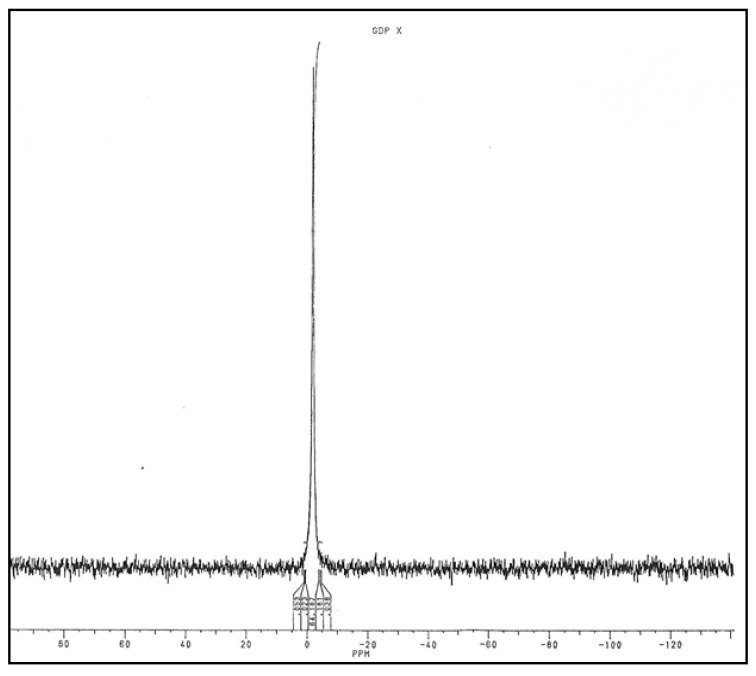
Analysis of Phosphorus Nuclear Magnetic Resonance of UV 260 nm-absorbing molecules released by GMD.

**Figure 12 antibiotics-12-01649-f012:**
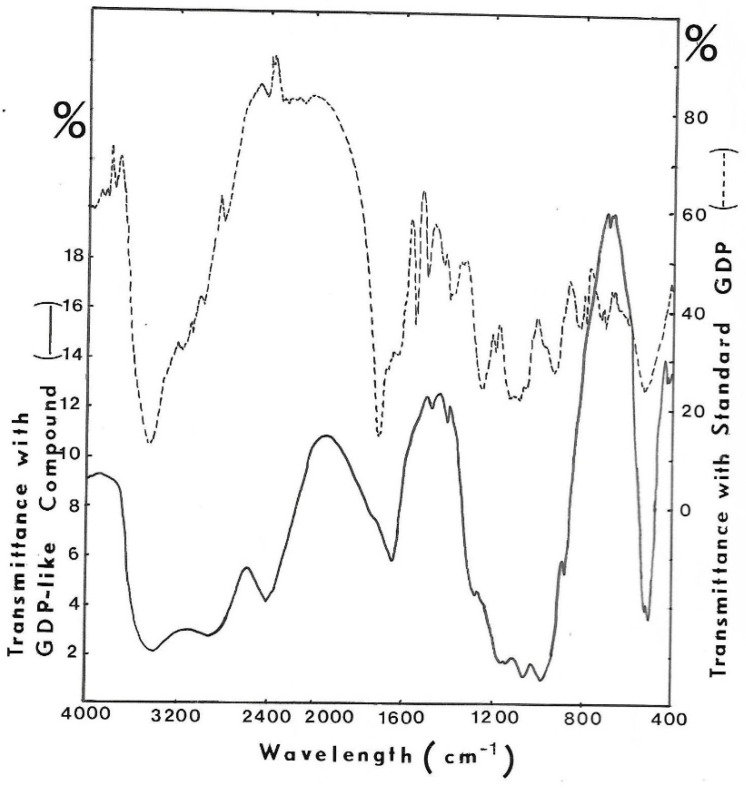
I.R. spectrum of purified UV-absorbing molecules released from GMD. KBr crystal was made with the GDP-like molecule harvested from GF05 gel exclusion chromatography, and the I.R. spectrum was recorded. KBr crystal was made with GDP to compare absorbing bands. (- - - -) I.R. spectrum of GDP, (−−−) I.R. spectrum of isolated compound.

**Figure 13 antibiotics-12-01649-f013:**
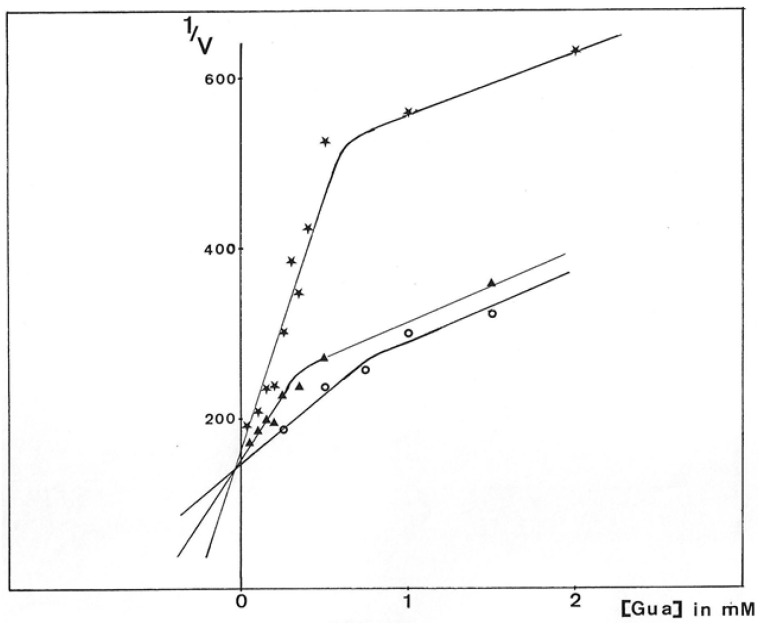
Dixon plot of *P. aeruginosa* NK125502 GMD inhibition by guanosine. GMD was incubated for 10 min at 37 °C with guanosine before the reaction was started by the addition of GDP-mannose and 1 mM NAD^+^. Initial rates were calculated from the slopes of the kinetics of NADH appearance. Results were expressed from the reciprocal analysis of Dixon, 1/V versus guanosine concentration in the reactional medium for different concentrations of substrate. 
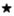
 0.15 mM GDP-mannose, 
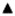
 0.25 mM GDP-mannose, and 
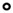
 0.5 mM GDP-mannose.

**Figure 14 antibiotics-12-01649-f014:**
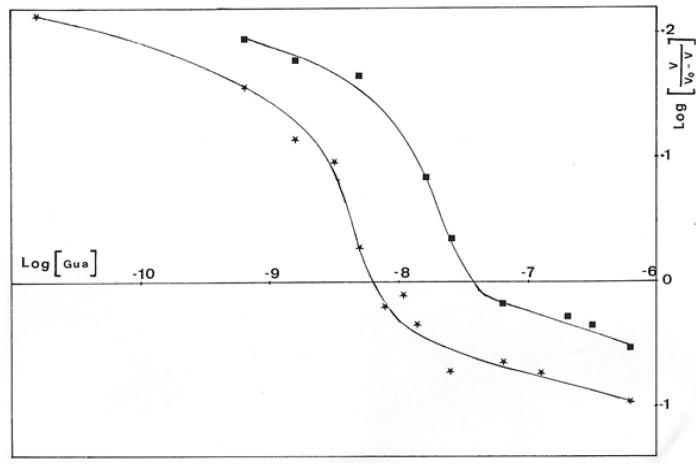
Hill’s representation of the inhibition of *P. aeruginosa* NK125502 GMD by guanosine. Initial rates of NADH appearance according to guanosine concentration in the incubation medium for different concentrations of substrate were plotted from an extension of the Hill’s equation: Log(I) versus Log (v/v_0_-v). 
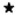
 0.15 mM GDP-mannose ; 
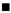
 0.2 mM GDP-mannose.

**Figure 15 antibiotics-12-01649-f015:**
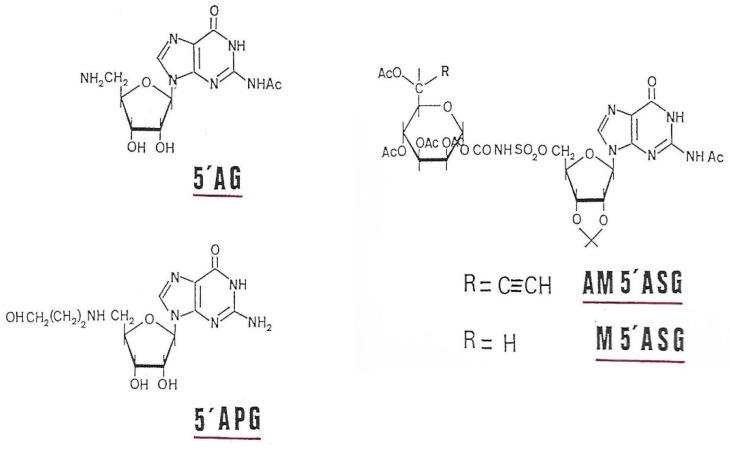
GMD inhibitors used in this study.

**Table 1 antibiotics-12-01649-t001:** Inhibition of GMD by guanosine, 5′aminoguanosine, and GDP and GDP-mannose derivatives.

Inhibitors	Percentage of GMD Inhibition at the Final Concentration of		
	0.1 mM	0.5 mM	1 mM
Mannose	0	0	0
Guanosine	3 ± 0.2	9 ± 2	20 ± 3
5′AG	18 ± 1	64 ± 3	75 ± 2
5′APG	33 ± 1	72 ± 2	85 ± 2
M5′ASG	62 ± 2	82 ± 2	92 ± 3
AM5′ASG	73 ± 2	90 ± 3	98 ± 2

## Data Availability

Data are contained within the article.

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
