# Peer review of "The GDP-Mannose Dehydrogenase of Pseudomonas aeruginosa: An Old and New Target to Fight against Antibiotics Resistance of Mucoid Strains"

_antibiotics, 2023, doi:10.3390/antibiotics12121649_

Round 1
Reviewer 1 Report
Comments and Suggestions for Authors
In the presented manuscript, the Author characterized one of the main enzymes involved in the synthesis of alginate -GDP-mannose dehydrogenase (GMD), in Pseudomonas aeruginosa. GMD catalyze double oxidation of GDP-mannose to GDP-mannuronate The results suggest that GMD acts as a dimer and exhibits strong negative cooperativity for its substrate. Oxidation occurs in two distinct steps with the substrate bound to the enzyme in the form of a hemiacetal. Interestingly, it was confirmed that GMD substrate analogues may effectively inhibit alginate production which may result in increased antibiotic susceptibility. I have the following comments:
In the presented work, the author characterized one of the main enzymes involved in the synthesis of alginate - GDP-mannose dehydrogenase (GMD) in Pseudomonas aeruginosa. GMDs catalyze the double oxidation of PKB-mannose to PKB-mannuronate. The results suggest that GMD acts as a dimer and exhibits strong negative cooperativity with its substrate. Oxidation occurs in two distinct steps in which the substrate is bound to the enzyme in the form of a hemiacetal. Interestingly, it was confirmed that GMD substrate analogues can effectively inhibit alginate production, which may result in increased sensitivity to antibiotics. I have the following comments:
Lines 129-131. Based on these results, it is difficult to confirm that there are two forms (monomer and dimer) of GMD. Gel filtration (Figure 3) was one of the initial stages of GMD purification. How pure was the preparation? (The author mentioned that GMD fractions were analyzed by SDS-PAGE, lines 108-109). To distinguish GMD monomers from dimers and assess their activity, samples obtained after Blue-Triactyl chromatography would need to be gel-filtered again.
Line 111 – The estimated pI of 6.8 differs from the theoretical pI (5.43) of The estimated pI of 6.8 differs from the theoretical pI (5.43) of GMD from P. aeruginosa PAO1 (Uniprot accession number: P11759 ALGD_PSEAE). Could you comment on this difference?
Fig. 4, 5, 7, 9 which of the dehydrogenases initially tested are shown in these graphs?
Fig. 1 –should be “Periplasm” instead of Periplasme
Line 536 - should be: [14]
The following articles could be added to the references:
doi: 10.1021/acschembio.0c00426.
doi: 10.3762/bjoc.18.142
Author Response
All comments have been taken into account and corrections made to the text.
Line 111. The difference in the pI values is difficult to explain. It may be linked to the different origins of the enzymes; algD gene has been cloned on a plasmid multicopy in E. coli and P. aeruginosa under the control of a tac promoter giving a hexamer of GMD. From mucoid strains of P. aeruginosa, GMD was found under monomeric and dimeric forms. This difference in native structure could perhaps explain the difference observed in pI values. As this pI value does not provide important information it has been deleted from the text.
Line 129-131. The monomer harboring the NAD binding site must adsorb onto the blue-Trisacryl column like the dimer. I think that a gel exclusion of the GMD eluted from the blue-Trisacryl chromatography should give the type of dispersion as the first exclusion gel.
The publications have been added to the references.
Reviewer 2 Report
Comments and Suggestions for Authors
The paper by Hulen describes the characterization of GMD and the use of inhibitors to block its activity. Overall, the paper is well organized and the experimental techniques are appropriate for the characterization. My comments are noted below.
1. The heart of the active site is the Cys at position 268. A mutation that changes this amino acid at this position would have been a good complement to showing this is part of the active site.
2. The Materials and Methods are missing some details that are important for reproducing the results. Line 505 How many rpms was the culture shaken? Line 510 What temperature was the culture grown in? Source for the antibiotics used? Source for the DNase used?
3. Gene names have the fourth letter capitalized, e.g. algD.
4. Line 29 Change to with the severe.
5. Line 31 add a hyphen after alginate.
6. Be consistent in either calling it epimerisation or epimerization, not use both.
7. Define your acronyms the first time you use them (e.g. CF, IR, NMR) and don't use an acronym but spell out if you use the term only once (e.g. IHF, CRP, IPTG).
8. Line 63 tac is italicized.
9. Line 65 Add a comma after polypeptide.
10. Units of measure are separated from the units themselves (e.g. 10 mM, pH 7, 1 g, etc.).
11. Line 97 subscript MgSO4, (NH4)2SO4.
12. Lines 108-111 The sentence rambles.
13. Line 114 Comma after filtration.
14. Line 126 Comma after chains.
15. Lines 132 - 135 Run-on sentence.
16. Line 138 Comma after extracts).
17. Line 158 Comma after aeruginosa.
18 Lines 172 - 173 binding, respectively,
19. Line 206 Bovine serum albumin is all lower case.
20. Line 208 GMD, the A280/260 ratio.
21. Line 287 Comma after diphosphate.
22. Line 334 antibiotic, such as tobramycin [39],
23. Lines 342 and 347 it is Table 1 not I.
24. Line 388 Comma after therefore.
25. Line 390 it is two not to.
26. Line 404 Space after [47].
27. What is adequate medium?
28. Line 423 rewrite the sentence.
29. Line 438 Change to two rather than 2.
30. Line 448 P. aeruginosa is italicized.
31. Line 468 Do you mean intermediate products?
32. Lines 474-476. Rewrite the sentence.
33. Lines 479-480 in vivo is italicized.
34. Line 493 from the Institut Pasteur collection was.
35. Line 519 30,000 x g.
36. Line 612. it is Lory not Lori.
Comments on the Quality of English Language
A number of English language changes need to be made, particularly the use of commas.
Author Response
1 The remplacement of Cys268 by a serine in the GMD polypeptide was carried out few years ago by site-directed mutagenesis. This change induced a loss of 95% of the activity (Roychoudhury et al., J. Biol. Chem., 267, p995).
2 Additional information regarding the products used and the temperature for bacterial growth have been added. Centrifugation speeds are expressed in "g" to be reproduced in any centrifuge.
All comments from n°3 to n° 36 have been taken into account and corrections made to the text.